# Clinical Evaluation of *Hovenia dulcis* Extract Combinations for Effective Hangover Relief in Humans

**DOI:** 10.3390/foods13244021

**Published:** 2024-12-12

**Authors:** Ki Won Lee, Guangpeng Xu, Dong Hyun Paik, Youn Young Shim, Martin J. T. Reaney, Ilbum Park, Sang-Hun Lee, Jong-Yul Park, Euddeum Park, Sung-Bum Lee, In Ah Kim, Ji Youn Hong, Young Jun Kim

**Affiliations:** 1Natural Products Convergence R&D Division, Kwangdong Pharmaceutical Co., Ltd., Gwacheon 13840, Republic of Koreapib975@ekdp.com (I.P.); spelljjt@ekdp.com (S.-H.L.); 11075@ekdp.com (J.-Y.P.); 13130@ekdp.com (E.P.); 2Department of Food and Biotechnology, Korea University, Sejong 30019, Republic of Korea; xuguangpeng@korea.ac.kr; 3Department of Food and Bioproduct Sciences, University of Saskatchewan, Saskatoon, SK S7N 5A8, Canada; martin.reaney@usask.ca; 4Prairie Tide Diversified Inc., Saskatoon, SK S7J 0R1, Canada; 5Department of Family Medicine, Soonchunhyang University Bucheon Hospital, Bucheon 22972, Republic of Korea; 6Global Medical Research Center, Seoul 06526, Republic of Korea; 7Department of Food Regulatory Science, Korea University, Sejong 30019, Republic of Korea; khjy1025@korea.ac.kr

**Keywords:** plant-based extract mixture, alcohol, hangover, *Hovenia dulcis* Thunb., *Pueraria lobata*, acute hangover scale, GC-MS, clinical trials

## Abstract

Alcohol consumption is associated with both short- and long-term adverse effects, including hangover symptoms. The objective of this study was to examine the potential benefits of traditional beverages containing a combination of *Hovenia dulcis* extract (HD) with either *Pueraria lobata* extract (HDPB) or glutathione yeast extract (HDGB) in abbreviating alcohol intoxication and mitigating hangover symptoms. A total of 25 participants between the ages of 19 and 40 who had previously experienced a hangover were evaluated in a randomized, double-blind, crossover, placebo (PLA)-controlled clinical trial. Results showed that lower blood alcohol concentrations in the HDPB and HDGB groups were significantly lower than in the PLA group at 0.25 and 0.5 h, suggesting that HD aids in early alcohol metabolism (0 h, *p* < 0.05). Analysis of the hourly Acute Hangover Scale (AHS) showed that all treatment groups had significantly reduced gastrointestinal disorder symptoms compared to the PLA group (*p* < 0.05). It can be confirmed that hangover symptoms can be significantly improved by consuming HD combination drinks, apart from the effect of reducing blood alcohol and acetaldehyde concentrations. Therefore, it is predicted that the consumption of natural phytochemicals added to HD is safe for humans and may help accelerate recovery from hangover symptoms.

## 1. Introduction

Excessive alcohol consumption is a widespread issue with significant social and health implications, contributing to approximately 3 million deaths worldwide each year and accounting for 5.1% of the global burden of disease and injury [1,2]. One of the most immediate and troubling consequences of alcohol consumption is a hangover, characterized by symptoms such as headache, nausea, fatigue, and dehydration [3,4]. These effects are largely caused by alcohol metabolism, which produces toxic intermediates such as acetaldehyde and induces oxidative stress in the liver [5]. Despite the numerous hangover remedies that have been proposed, their scientific validation remains limited [6].

Aspirin and acetaminophen are commonly utilized in the management of post-alcohol consumption headaches. However, these commonly used painkillers have not been subjected to rigorous evaluation in clinical trials specifically designed to assess their efficacy in treating hangovers [7]. Furthermore, pharmacological agents such as disulfiram [an aldehyde dehydrogenase (ALDH) inhibitor], naltrexone (an opioid antagonist), and acamprosate (an NMDA/glutamate receptor modulator) are employed in the treatment of alcohol abuse. However, these agents frequently induce considerable adverse effects, including ataxia and confusion [8,9,10,11]. Consequently, there is an increasing requirement for natural remedies that can alleviate the symptoms of a hangover and protect liver health. Recently, plant-based extract mixtures have attracted attention as a potential source of benefits in alleviating hangover symptoms and promoting liver health [12].

*Hovenia dulcis* fruit extract (HD), used in traditional East Asian medicine, is known for its hepatoprotective properties [13,14]. HD is also known as the oriental raisin tree. It has been demonstrated to support liver detoxification by enhancing the activity of enzymes that break down acetaldehyde, reducing liver inflammation, and mitigating oxidative stress [15,16]. One of its key bioactive compounds, dihydromyricetin (DHM), promotes the breakdown of alcohol byproducts and helps prevent alcohol-induced liver damage [17,18]. Other previous studies have also found that in the fight against acute alcoholic liver damage, HD works by reducing the triglyceride levels in mouse livers [19], while in chronic alcoholic fatty liver damage, HD can reduce triglyceride accumulation in the liver and achieve the effect of protecting liver tissue [20].

*Pueraria lobata* root extract (PL), or kudzu root, is another traditional remedy for alcohol-related conditions [21]. PL is a rich source of isoflavones, including puerarin, daidzin, and daidzein. These isoflavones have been demonstrated to modulate alcohol metabolism by inhibiting the enzymes alcohol dehydrogenase (ADH) and ALDH, which are responsible for the metabolism of alcohol and acetaldehyde [22,23]. This helps reduce acetaldehyde accumulation and prevents oxidative damage to liver cells. In addition, PL has been shown to improve blood circulation and relieve symptoms associated with vasodilation, such as headache [24]. While some studies report its inhibitory effects on ADH activity, others suggest antioxidant properties that may indirectly improve alcohol clearance and protect liver cells from oxidative damage [25].

Glutathione-enriched yeast extract (GY) is another promising ingredient for hangover relief due to its potent antioxidant properties [26,27]. Glutathione is a tripeptide that is central in neutralizing reactive oxygen species (ROS) and detoxifying harmful substances in the liver [28]. By enhancing endogenous glutathione levels, this extract accelerates the breakdown of acetaldehyde, reduces oxidative stress, and protects liver cells from damage [29].

This trial evaluates the efficacy of HD, PL, and GY in combination to prevent hangover symptoms through a randomized, double-blind, placebo (PLA)-controlled, crossover clinical trial. By analyzing blood alcohol and acetaldehyde levels post-alcohol consumption, we aim to provide robust evidence to support the use of these extracts as functional ingredients. This research not only bridges the gap in the clinical validation of natural remedies but also highlights their potential for the development of effective functional foods to mitigate alcohol-related harm.

## 2. Materials and Methods

### 2.1. Clinical Trial Design

This clinical trial was approved by the Institutional Review Board of the Global Medical Research Center (IRB No. GIRB-24216-ZM) and was registered with the Clinical Research Information Service (CRIS) of the Centers for Disease Control and Prevention (CRIS No. KCT0009900). Conducted from 29 April to 16 June 2024, the trial followed a randomized, double-blind, parallel-group, PLA-controlled design. On the screening day, participants were randomized into the PLA, HDPB, and HDGB groups and administered treatments following a double-blind protocol (Figure 1). The randomization sequence was computer-generated by a statistician working with the study sponsor. Both participants and researchers remained blinded to treatment assignments until the study’s conclusion. A priori power analysis determined the required sample size to evaluate treatment efficacy [30].

The trial used a crossover design with participants randomized to one of six groups, with a target of 30 participants to allow for a potential 20% dropout rate (Table 1). On visit days 1 and 2 (Day 0), participants underwent screenings to verify eligibility based on demographic, lifestyle, medical, and medication history, as well as physical assessments like blood pressure, pulse, anthropometry (height, BMI, and weight), clinical pathology, and a pregnancy test for women of childbearing age. A drinking behavior questionnaire gathered information on alcohol type, quantity, frequency, and recent hangover occurrences. At Visit 2 (Day 0), eligible participants were assigned to one of the study’s six arms using a block randomization method to maintain a balanced distribution across groups in a 1:1:1:1:1:1 ratio. Participants received a single dose of the designated investigational food (Test Food I, Test Food II, Test Food III, or PLA) on Visits 2, 3, and 4, with a 7-day washout period (±7 days) between each visit (Figure 1).

### 2.2. Participants

Participants for this clinical study were recruited from the Global Medical Research Center in Seoul (Republic of Korea). Eligibility criteria included the following: (1) age between 19 and 40 years, (2) a body mass index (BMI) from 18.5 to 25 kg/m^2^, and (3) a history of experiencing hangovers. In addition, participants had to fulfill the following conditions: (1) alcohol consumption within the last 30 days, (2) an exhaled alcohol concentration of 0.00% at the initial visit on Day 2, and (3) consent to join the clinical trial and sign an informed consent form. It is of particular importance to note that the hangover history criterion required participants to have previously experienced symptoms such as headaches, nausea, fatigue, and sensitivity to light and sound. This ensured that they were suitable for the evaluation of hangover mitigation techniques.

Exclusion criteria were set to ensure participant safety and data integrity. The screening criteria were adjusted based on Shin [31]. Excluded individuals included those who (1) were receiving treatment for significant medical conditions, such as cardiovascular, immune, respiratory, endocrine (e.g., diabetes), renal, neurologic, psychiatric, infectious diseases, or cancer; (2) had a history of peptic ulcer disease, reflux esophagitis, or severe gastrointestinal issues; (3) were pregnant, lactating, or planning pregnancy during the study; (4) had an alcohol use disorder or recent excessive alcohol consumption that might interfere with the study; (5) were taking medications likely to impact alcohol metabolism (e.g., antidepressants) or with a bleeding risk (e.g., warfarin and aspirin); (6) had used medications to enhance liver function (e.g., disulfiram) or those affecting drug-metabolizing enzymes within a specified period before screening; (7) had abnormal lab values for AST (glutamic oxaloacetic transaminase, GOT), ALT (glutamic pyruvate transaminase, GPT), creatinine, thyroid-stimulating hormone, or fasting glucose; (8) had participated in another interventional clinical trial in the previous month or planned to do so during this study; (9) had a known allergy or sensitivity to any trial ingredients; and (10) were otherwise deemed unsuitable by the investigator’s assessment.

### 2.3. Procedures

Kwangdong Pharmaceutical Co. Ltd. (Gwacheon, Republic of Korea) generously provided concentrated aqueous samples. Three formulations were administered orally as 500 mL beverages per bottle, including a control (PLA) (Table 2): 0.731% (*v*/*v*) *H. dulcis* extract (HD), HD combined with 0.1% (*v*/*v*) PL (HDPB), and HD combined with 0.02% (*w*/*v*) GY (HDGB). The control sample (PLA) contained 0.021% (*w*/*v*) caramel pigment powder and 0.11% (*w*/*v*) flavors, with the remaining volume comprising purified water. Sub-ingredients included 0.28% (*w*/*v*) natural plant and grain concentrates (such as *Acer tegmentosum* Maxim., *Alnus japonica*, rice embryo complex fermented extract, and albumin), 0.020% vitamin C, 0.029% sodium bicarbonate, 0.012% glycine, and 0.14% flavoring agents as minor components (*w*/*v*). All samples were indistinguishable in appearance, taste, sweetness, color, packaging, and additional ingredients.

To evaluate the efficacy and safety of these natural plant and fermented extracts in alleviating hangover symptoms, the potential hangover-relieving efficacy of HD, PL, and GY was first verified in Sprague Dawley male rats in preparation for subsequent clinical trials. After confirming the efficacy of each extract individually, a randomized, double-blind, PLA-controlled, crossover clinical trial was conducted. This study compared the effects of natural plant and fermented extract combinations and PLA on hangover symptoms and liver function. Participants were given a combination of HD and PL, a combination of HD and GY, or PLA. The crossover design reduced individual variability by allowing each participant to serve as their own control [32]. The double-blind design prevented bias from both participants and researchers [33], and the PLA group helped verify the efficacy of the treatment [34].

All participants were instructed to abstain from alcohol for 24 h prior to each visit [Visit 1, Visit 2 (Day 1), and Visit 3 (Day 1)]. They were then provided with an identical meal and subsequently ingested either the PLA or the intervention beverage 2 h later. Thirty minutes post-consumption of the test beverage, participants consumed whiskey (0.9 g/kg body weight) with 40% alcohol in two portions over a 30 min interval. Participants were instructed to fast following alcohol consumption. On Visit 1 and subsequent Visits 2, 3, and 4 (Day 1), subjects were required to abstain from alcohol for at least 24 h before the next visit. The following morning, approximately 15 h after alcohol ingestion, participants completed a hangover symptom questionnaire upon waking.

### 2.4. Outcome Measures

#### 2.4.1. Acute Hangover Scale (AHS)

In accordance with the methodology proposed by Verster et al. [35], the Acute Hangover Scale (AHS) was employed for the assessment of hangover symptoms, comprising nine items that evaluate both individual symptoms and overall severity. Each item is rated on a 0 to 7 scale, with the total AHS score representing the average of all items. Symptom levels are categorized as ‘none’ (0 points), ‘mild’ (1 point), ‘moderate’ (4 points), and ‘incapacitating’ (7 points).

#### 2.4.2. Alcohol and Acetaldehyde Analysis

Blood samples were collected at intervals of 0, 0.25, 0.5, 1, 2, 4, 6, and 15 h following alcohol intake on Visits 2, 3, and 4 (Days 0 and 1). For sample collection, a catheter primed with saline was inserted into a vein in the participant’s arm, drawing 5 mL of blood each time. Samples were stored in BD Vacutainer^®^ NaF tubes at 4 °C and analyzed immediately. Blood handling and analysis followed the protocols outlined in the laboratory manual, and samples were discarded upon completion of the analysis.

#### 2.4.3. Analysis of Alcohol and Acetaldehyde in Blood

Blood alcohol and acetaldehyde content were measured by gas chromatography–mass spectrometry (GC-MS) according to a standardized protocol [36]. Clinical blood samples were maintained on ice throughout the analysis. For each assay, 200 µL of plasma was mixed with 500 µL of saturated NaCl solution and 100 µL of 0.005% *n*-butanol, which served as an internal standard, in a headspace vial. Analysis was performed using an Agilent 5977 series GC system (Agilent Technologies, Palo Alto, CA, USA) paired with a CTC headspace GC/MS detector (CTC Analytics AG, Zwingen, Switzerland). Headspace sample equilibration was performed at 70 °C for 10 min. A sample injection volume of 250 µL was used with a split ratio of 100:1. Ethanol, acetaldehyde, and *n*-butanol were separated on a Discovery HP-INNOWAX column (0.32 mm × 30 m, 0.5 µm, Sigma-Aldrich, St. Louis, MO, USA) employing a helium gas carrier at a flow rate of 3 mL/min, with the interfacial temperature set at 200 °C. The GC oven temperature program began at 35 °C (held for 3 min), was ramped to 85 °C at 40 °C/min, and then maintained for an additional 2 min. The mass spectrometer was operated in single ion monitoring mode, with target *m*/*z* values of 45, 46, and 31 for ethanol and 43, 41, and 29 for acetaldehyde, with quantification based on *m*/*z* 45 for ethanol and *m*/*z* 43 for acetaldehyde, respectively.

### 2.5. Safety Assessments

Vital signs, including blood pressure (systolic and diastolic), temperature, pulse rate, and clinical laboratory tests (hematology, biochemistry, and urinalysis), were assessed at screening (Visit 1) and again on Day 1 (Visit 2). Adverse events were monitored and documented through participant interviews and questionnaires throughout the trial. 

### 2.6. ADH and ALDH Enzyme Activity Analysis 

Markers of alcohol hangover were evaluated by measuring ADH and ALDH enzyme activities in plasma using commercial kits for ADH (K787-100; BioVision Inc., Milpitas, CA, USA) and ALDH (K731-100; BioVision Inc.) according to the manufacturer’s protocols.

### 2.7. Statistical Analysis

Statistical analyses were conducted with SAS software (version 9.4, SAS Institute, Cary, NC, USA). Descriptive statistics were used to calculate means and standard deviation (SD), with significance determined with two-tailed tests at a threshold of *p* < 0.05. A one-way repeated measures analysis of variance (RM ANOVA) was used in comparisons of blood alcohol and acetaldehyde levels (including hourly concentrations, C_max_, T_max_, and AUC) to identify significant differences in intake effects between groups. An additional ANOVA was used to test for between-group effects under the assumption of no carryover effect. Changes in AHS scores over time were also assessed to determine statistical significance. All adverse event rates were compared between groups using Chi-square or Fisher’s exact tests. Hematology and blood chemistry values between groups (Test Food I vs. control, Test Food II vs. control, and Test Food III vs. control) were analyzed for statistically significant differences using ANOVA or the Kruskal–Wallis test based on normality, with post hoc analysis as necessary. Urinalysis results, both normal and abnormal, were compared by Chi-square or Fisher’s exact tests. Within the group, pre- and post-consumption changes in blood pressure and pulse rate were analyzed using paired *t*-tests, while between-group comparisons were assessed using ANOVA or the Kruskal–Wallis test based on data distribution. 

## 3. Results and Discussion

### 3.1. Enrollment

The enrollment criteria and participant flow help ensure the reliability of this study by maintaining a well-defined sample, which is critical for interpreting outcomes accurately [37]. This study enrolled 30 participants, with 4 excluded during the screening process. From 3 April 2024 to 25 May 2024, 25 participants proceeded to the experimental analysis phase. Five participants were excluded during the experimental period. To analyze efficacy, the pre-protocol (PP) set was the primary analysis method, with the full analysis (FA) set used as a supplement. Demographic data analysis was also based on the PP set, while the safety data relied on the Safety Set. In this study, no exclusions were made from the PP set among participants in the FA set, ensuring both FA and PP analyses included 25 cases (*n* = 25) (Figure 2). The careful exclusion and set assignment were aimed at minimizing bias and enhancing the validity of the analysis.

### 3.2. General Participant Characteristics

Ensuring baseline comparability between groups is crucial in clinical trials because it allows researchers to accurately isolate and assess the effect of the intervention on the outcome of interest, thereby increasing the internal validity of the trial [38]. In a randomized controlled trial, comparable baseline characteristics between groups ensure that any observed differences in outcomes can be confidently attributed to the intervention, minimizing the risk of confounding factors. This strengthens causal inferences and reduces selection bias, thereby increasing the internal and external validity of the trial [39,40]. Table 3 presents the participants’ demographics and baseline characteristics, analyzed to identify differences between the intake groups. The study group comprised 14 males (56%) and 11 females (44%), with no statistically significant differences in baseline characteristics, such as age, exercise habits, alcohol intake, smoking status, sleep duration, height, and weight. This homogeneity in baseline characteristics suggests a balanced allocation across groups, which reduces potential confounding effects and strengthens the integrity of the comparative analysis.

### 3.3. Biochemical Parameters

The lack of significant changes in biochemical parameters, as shown by the hematological and vital sign stability in both test and control groups, underscores the safety profile of the intervention even after alcohol consumption [41]. This is a critical component for advancing this trial towards larger, more diverse populations [42,43]. Safety evaluation was conducted using a Safety Set analysis on the 30 subjects randomized to both test and control groups. The results revealed no significant differences in hematological parameters after a single day of beverage consumption (Table 4). Additionally, no significant differences were observed at 15 h post-drinking across all hematologic tests (Table 4). While some biochemical markers, such as AST and bilirubin, had lower *p* values, they did not reach statistical significance (*p* > 0.05), implying no evident safety risks. In addition, there was no statistically significant difference between the groups in any of the urine test parameters, including glucose, ketones, and bilirubin, 15 h after drinking. Therefore, no safety concerns were reported.

Table 5 shows the results of vital signs (blood pressure and pulse) measured at baseline and 15 h after completion of drinking, analyzed in the Safety Set. There were no statistically significant differences in vital sign (blood pressure and pulse) changes 15 h after completion of drinking between the intake groups. The evaluation, particularly at 15 h post-intervention, reinforces the confidence that the intervention’s biochemical and vital sign stability is not only consistent immediately after consumption but also sustainable, an important factor for progressing clinical trials [44].

### 3.4. Survey of Hangover Symptoms

The AHS scores were compared between the experimental and control groups to determine any statistically significant differences in hangover symptoms. Table 6 shows the AHS results measured 15 h after alcohol consumption. There was a statistically significant reduction in the AHS gastrointestinal disorder score 15 h after completion of drinking in the HDPB and HDGB groups compared to the PLA group (*p* < 0.05). There was no statistical significance in any of the other efficacy endpoints. The AHS gastrointestinal disturbance score at 15 h after completion of alcohol drinking by GG genotype was 0.18 ± 0.50 in the HDPB group, 0.23 ± 0.69 in the HDGB group, and 0.50 ± 1.01 in the PLA group, with statistically significant differences between the groups (*p* < 0.05) (Table 7). Post hoc tests using Dunnett’s test showed a significant difference between HDPB and the control group (*p* < 0.05) and no significant difference between HDGB and PLA. These findings suggest that HDPB may effectively reduce gastrointestinal hangover symptoms, a finding that could be valuable for developing targeted hangover relief solutions. Previous studies have shown that *HD* and *PL* extracts have a synergistic effect in treating intestinal inflammation and dysfunction and can significantly reduce stress response [45,46]. Wei et al. studied the chemical components of PL and HD extracts using high-performance liquid chromatography and Fourier transform ion cyclotron resonance mass spectrometry and identified 48 chemical components [47], including genistein, with potential beneficial effects (including antioxidant, hepatoprotective, anticancer, and cardioprotective effects) and kaempferol [48], a flavonoid compound with anti-inflammatory properties [49]. These plant-derived bioactive compounds have a strong protective ability in alleviating intestinal abnormalities [50]. Although no significant differences were observed in other measures, the scores of the HDPB group were consistently lower than those of the PLA group. This indirectly demonstrates the efficacy of the mixed extract of HD and PL in alleviating hangover symptoms.

The area under the concentration–time curve (AUC) was also statistically significantly larger in the HDPB group than in the PLA group. The area under the incremental AUC (iAUC) was statistically significantly larger in the HDPB group compared to the PLA group (*p* < 0.05) (Table 8).

### 3.5. Changes in Blood Alcohol and Acetaldehyde Levels

Figure 3 shows the results of measuring the change in blood alcohol and acetaldehyde concentrations from 0 to 15 h after drinking. At 0 h (*p* < 0.05), 0.25 h, and 0.50 h, the HDPB and HDGB groups had lower blood alcohol levels than PLA, showing that HD slows alcohol absorption early. By 6 h, HDGB had lower blood alcohol levels (*p* < 0.05) than HDPB, suggesting a synergistic effect between HD and GY in accelerating alcohol metabolism (Figure 3A). This is consistent with the intestinal protective properties of HD [51]. Similarly, the flavonoid DHM in HD has been shown to significantly reduce alcohol digestibility in rats under voluntary drinking conditions [46]. Furthermore, blood acetaldehyde levels at 6 h were significantly higher in the HDPB and HDGB groups than in the PLA group, suggesting increased efficiency of alcohol catabolism by HD (Figure 3B). Similarly, recent studies have shown that HD increases ADH activity [52], which accelerates the conversion of blood ethanol to acetaldehyde [53]. However, the accumulation of acetaldehyde may be due to limited ALDH activity, resulting in slower acetaldehyde clearance, an effect consistent with our observations. The accelerated reduction in alcohol levels in the HDPB and HDGB groups highlights the potential efficacy of these formulations in promoting faster alcohol metabolism, which may be key to reducing hangover severity.

Interestingly, it has been reported that PL exerts its effects through a dual role in alcohol metabolism. First, isoflavones, such as puerarin, can inhibit ADH, potentially reducing acetaldehyde accumulation and preventing oxidative damage to liver cells [54]. Second, under certain conditions, PL extract may indirectly enhance alcohol metabolism by reducing oxidative stress and supporting overall enzymatic activity [55]. This apparent discrepancy may be due to differences in experimental models, dosages, or the specific composition of the extracts used. Our results suggest that PL at a concentration of 0.1% (*w*/*v*) may exert its effects primarily through antioxidant pathways rather than direct inhibition of ADH. This hypothesis is supported by the observed reduction in blood alcohol levels after 6 h, together with increased acetaldehyde concentrations, which is consistent with a partial enhancement of ADH activity in vivo.

## 4. Conclusions

A hangover is an uncomfortable physiological symptom induced by drinking alcohol that is caused by acetaldehyde, a toxic substance produced by ADH metabolism. Rapid breakdown of alcohol and acetaldehyde is essential to mitigate symptoms after drinking. This human application study evaluated the efficacy and safety of consuming HD combined with PL (HDPB) or HD combined with GY (HDGB). This study focused on reducing alcohol concentration and improving hangover symptoms in people aged 19 to 40 years who have experienced a hangover after drinking alcohol. Twenty-five participants who met the eligibility criteria were recruited for a double-blind, randomized, PLA-controlled crossover trial. As a result, the groups that showed a statistically significant reduction in gastrointestinal disorders, which may be related to changes in the gut microbiome post-drinking, were the HDPB and HDGB groups compared to the PLA group. Although not statistically significant, the HDPB group had lower scores than the PLA group at all post-drinking time points, including in thirst and anorexia. This suggests a reduction in the incidence of hangover symptoms after HDPB consumption. As there was no correlation between the occurrence of other hangover symptoms and alcohol concentration, it can be confirmed that, apart from the effect of reducing alcohol concentration, hangover symptoms can be significantly improved by consuming the test food. Therefore, it is predicted that the consumption of natural phytochemicals added to HD is safe for humans and may help to accelerate recovery from hangover symptoms. Therefore, it is predicted that the consumption of natural phytochemicals added to HD is safe for the human body and may help speed up the recovery from hangover symptoms, and the underlying mechanisms of these effects will be investigated in more detail in future experiments.

Given the safety and efficacy demonstrated in this clinical trial, these findings provide a basis for the development of functional foods or dietary supplements targeting hangover relief. The integration of HD with other extracts, such as PL and GY, offers a promising avenue for the creation of health-promoting products. Future research should explore these formulations in larger, more diverse populations to confirm these effects and facilitate their commercial application. Such advances could significantly reduce the societal and individual harm associated with excessive alcohol consumption, in line with the growing demand for safe, plant-based solutions in the functional food and nutraceutical markets.

## Figures and Tables

**Figure 1 foods-13-04021-f001:**
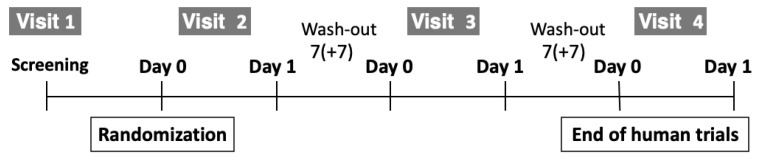
Overview of human clinical trials.

**Figure 2 foods-13-04021-f002:**
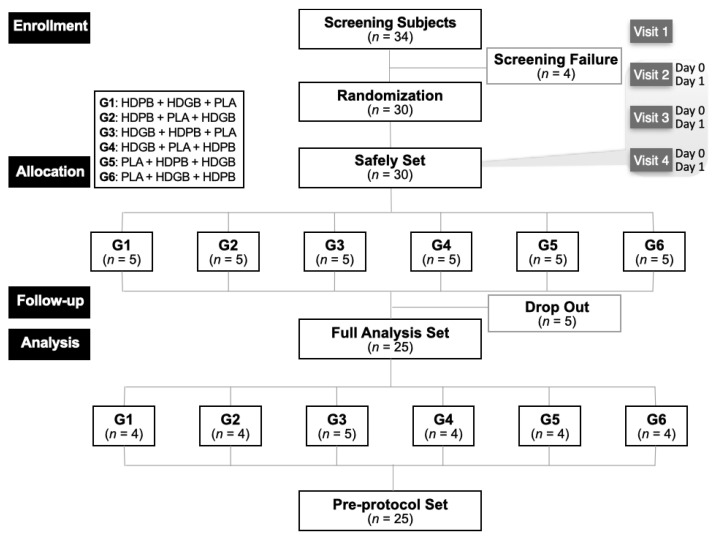
Flowchart showing study participant selection and assignment. Thirty eligible subjects were randomly assigned to one of the six groups, followed by a one-week washout period and subsequent crossover to the alternate group. All subjects completed the study. HDPB: HD combined with 0.1% *P. lobata* extract, HDGB: HD combined with 0.02% glutathione yeast extract, PLA: placebo, G#: group no.

**Figure 3 foods-13-04021-f003:**
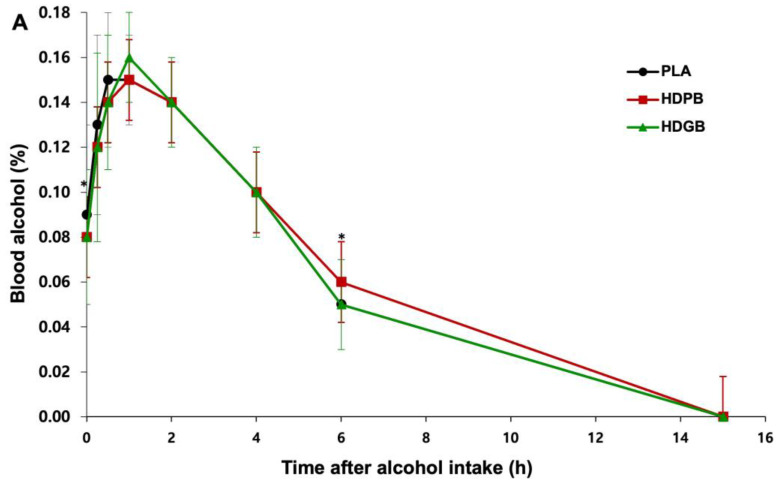
Effect of samples and PLA on (**A**) blood alcohol and (**B**) acetaldehyde concentrations after alcohol consumption. Values are presented as mean ± SD. Compared within groups; *p* value for RM ANOVA. * *p* < 0.05.

**Table 1 foods-13-04021-t001:** Timeline for human clinical trials.

Period		Screening ^1^	Active Treatment ^2^							
**Visit**		**1**	**2**		**3**		**4**		**5**	
**Day**		**–14**	**0**	**1**	**0**	**1**	**0**	**1**	**0**	**1**
**Window Period (Day)** ** ^3^ **					**+7**		**+7**		**+7**	
Written consent		✓								
Demographics ^4^		✓								
Lifestyle research ^5^		✓								
Medical and surgical history ^6^		✓	✓							
Medication history, non-medication history ^7^		✓	✓	✓	✓	✓	✓	✓	✓	✓
Physical examination		✓	✓	✓	✓	✓	✓	✓	✓	✓
Vital signs (blood pressure, pulse) ^8^		✓	✓	✓	✓	✓	✓	✓	✓	✓
Body instrumentation ^9^	Height and BMI	✓								
	Weight	✓	✓	✓		✓		✓		✓
Clinical pathology ^10^		✓		✓		✓		✓		✓
Pregnancy reaction test ^10^		✓								
Alcohol degradation genetic testing ^11^		✓								
Breath alcohol test ^12^			✓		✓		✓		✓	
Drinking Habits Survey		✓	✓		✓		✓		✓	
Validity evaluation	Blood alcohol, acetaldehyde concentrations ^13^		✓	✓	✓	✓	✓	✓	✓	✓
	Alcohol hangover scale (AHS) ^14^		✓	✓	✓	✓	✓	✓	✓	✓
Evaluating human subject suitability		✓	✓							
Randomization			✓							
Consumption of human investigational foods/alcohol ^15^			✓		✓		✓		✓	
Checking for adverse events			✓	✓	✓	✓	✓	✓	✓	✓
Human subject training ^16^		✓		✓		✓		✓		

^1^ Visit 2 (Day 0) must occur within 14 days of Visit 1. Visit 1 and Visit 2 (Day 0) must occur on the same day. If some tests are missing from Visit 1, they can be performed before randomization to Visit 2 (Day 0). ^2^ Visits 2, 3, and 4 will take place overnight at the site and will include the same diet 2 h prior to the ingestion of the investigational food. ^3^ Visits 3 and 4 (Day 0) should be performed after a rest period of 7 days after the fallow period. ^4^ On Visit 1, ask for gender, date of birth, and age. ^5^ At Visit 1, ask about smoking, exercise, and total sleep duration. ^6^ A review of the medication and non-medication history within a 30-day period preceding Visit 1 was conducted. Subsequently, at each visit, an examination of changes in the medication and non-medication history in comparison to those identified at Visit 1 was performed and documented. ^7^ Medication and non-medication history within 1 month (30 days) of Visit 1 was investigated. At each subsequent visit, changes in medication and non-medication history compared to those identified at Visit 1 were investigated and recorded. ^8^ On Visits 2, 3, and 4 (Days 0 and 1), vital signs (blood pressure and pulse) were measured 2 h prior to ingestion of the investigational food and 15 h after completion of drinking. ^9^ Height was measured to the nearest 0.1 cm and weight was rounded to the nearest 0.1 kg. ^10^ Human clinical trial subjects were fasted for 8 h prior to blood draws and were screened for the following items: Clinical pathology tests at Visit 1 were applicable if results were available within 4 weeks before Visit 1 (excluding pregnancy reactivity tests) and could be re-tested for abnormal results at the discretion of the human clinical trial investigator. Clinicopathologic examinations on Visits 2, 3, and 4 (Day 1) were performed 15 h after completion of drinking. Specimens were stored and analyzed by the external laboratory manual and discarded immediately after analysis without further use. Hematologic tests: WBC, RBC, Hb, Hct, platelet, neutrophil, lymphocyte, monocyte, eosinophil, and basophil. Blood chemistry tests: AST (GOT), ALT (GPT), γ-GTP, Total protein, Blood Urea Nitrogen (BUN), creatinine, Uric acid, Alkaline Phosphatase (ALP), bilirubin, glucose, Total cholesterol, HDL cholesterol, LDL cholesterol, and triglyceride. Urinalysis: pH, protein, glucose, ketone, bilirubin, RBC (Erythrocyte), Urobilinogen, Nitrite, WBC (Leukocyte). ^11^ Specimens will be stored and analyzed by the Exotic Petrographic Institution Manual and discarded immediately after completion of the analysis and not used for secondary purposes ^12^ before randomization at Visit 2 (Day 0) and before the start of the human clinical trial at Visits 3 and 4 (Day 0) to determine alcohol consumption. ^13^ Samples were collected on Visits 2, 3, and 4 (Days 0 and 1) before drinking and at 0, 0.25, 0.5, 1, 2, 4, 6, and 15 h after completion of drinking. Specimens were stored and analyzed according to the external laboratory manual and discarded immediately after analysis without further use for secondary purposes. ^14^ Visits 2, 3, and 4 (Days 0 and 1) were conducted 1, 2, 4, and 15 h after completion of drinking. ^15^ Human food was consumed in divided portions at 30 min intervals before alcohol consumption, followed by the ingestion of both human food and alcohol (0.9 g/kg bw) together in divided portions at 30 min intervals. Fasting was observed after alcohol consumption. ^16^ Human subjects were instructed to abstain from alcohol within 24 h of the next visit on Visit 1 and Visits 2 and 3 (Day 1).

**Table 2 foods-13-04021-t002:** Ingredients of the PLA group and two experimental beverage groups.

Ingredients (g)		Beverage Group Content (%)		
		HDPB	HDGB	PLA
Main ingredients	HD ^1^	0.731	0.731	0
	PL ^2^	0.100	0.000	0
	GY ^3^	0.000	0.020	0
Sub-ingredients	*Acer tegmentosum* Maximowoc	0.100	0.100	0
	*Alnus japonica*	0.100	0.100	0
	Rice embryo complex fermented extract	0.010	0.010	0
	Albumin	0.070	0.070	0
	Vitamin C	0.020	0.020	0
	Sodium bicarbonate	0.029	0.029	0
	Glycine	0.012	0.012	0
Other ingredents (excipients)	Caramel pigment powder	0.000	0.000	0.021
	Flavors	0.140	0.140	0.110
	Purified water	98.688	98.768	99.869

Abbreviations: HDPB: HD beverage combined with 0.1% PL; HDGB: HD beverage combined with 0.02% GY; PLA: placebo; HD, *H. dulcis* extract; PL, *P. lobata* extract; GY, glutathione yeast extract. ^1^ Concentrated aqueous HD: 60 Brix, 50% solid on drying. ^2^ Concentrated aqueous PL: 50 Brix, 40% solid on drying. ^3^ Powder GY: yeast extract 100% (*L*-glutathione > 10%).

**Table 3 foods-13-04021-t003:** Demographic information and pre-intake characteristics of the participants (PP set ^1^).

Variables		G1 (*n =* 4)	G2 (*n =* 4)	G3 (*n =* 5)	G4 (*n =* 4)	G5 (*n =* 4)	G6 (*n =* 4)	*p* Value ^2^
Gender *n* (%)	Male	2 (50.0)	3 (75.0)	3 (60.0)	2 (50.0)	2 (50.0)	2 (50.0)	1.0000 (F)
	Female	2 (50.0)	1 (25.0)	2 (40.0)	2 (50.0)	2 (50.0)	2 (50.0)	
Age	Mean ± SD	27.5 ± 6.6	27.0 ± 5.9	29.6 ± 3.9	31.0 ± 6.9	30.5 ± 6.0	29.0 ± 5.9	0.9033 (A)
	Min, Max	20.0, 36.0	21.0, 35.0	25.0, 34.0	24.0, 40.0	24.0, 38.0	21.0, 35.0	
Whether you smoke, *n* (%)	Yes	1 (25.0)	2 (50.0)	3 (60.0)	0 (0.0)	1 (25.0)	3 (75.0)	0.3341 (F)
	No	3 (75.0)	2 (50.0)	2 (40.0)	4 (10.0)	3 (75.0)	1 (25.0)	
Smoking amount (cigarette/day)	Mean ± SD	10.0	7.00 ± 4.24	8.67 ± 2.31	NS ^3^	10.0	6.67 ± 3.51	0.7163 (K)
	Min, Max	10.0, 10.0	4.0, 10.0	6.0, 10.0	NS	10.0, 10.0	3.0, 10.0	
Exercise or not *n* (%)	No	1 (25.0)	1 (25.0)	2 (40.0)	1 (25.0)	1 (25.0)	2 (50.0)	0.2821 (F)
	1–2 times/week	2 (50.0)	0 (0.0)	2 (40.0)	3 (75.0)	2 (50.0)	0 (0.0)	
	3–4 times/week	0 (0.0)	3 (75.0)	0 (0.0)	0 (0.0)	1 (25.0)	2 (50.0)	
	5–6 times/week	1 (25.0)	0 (0.0)	1 (20.0)	0 (0.0)	0 (0.0)	0 (0.0)	
	Daily	0 (0.0)	0 (0.0)	0 (0.0)	0 (0.0)	0 (0.0)	0 (0.0)	
Total sleep time (h/day)	Mean ± SD	8.75 ± 0.96	7.00 ± 0.82	7.20 ± 1.30	6.50 ± 0.58	6.75 ± 1.26	6.63 ± 0.48	0.1065 (K)
	Min, Max	8.0, 10.0	6.0, 8.0	6.0, 9.0	6.0, 7.0	5.0, 8.0	6.0, 7.0	
Height (cm)	Mean ± SD	164.7 ± 8.1	173.3 ± 8.7	169.2 ± 9.1	166.6 ± 8.3	166.9 ± 6.6	168.5 ± 6.6	0.7378 (A)
	Min, Max	154.9, 171.5	161.1, 180.7	156.5, 180.2	158.1, 175.5	160.0, 175.0	159.8, 174.7	
Weight (kg)	Mean ± SD Min, Max	58.9 ± 2.6	63.5 ± 9.6	64.1 ± 6.8	58.8 ± 9.6	60.4 ± 5.1	66.5 ± 3.4	0.5015 (A)
		55.5, 61.4	52.3, 75.5	54.0, 70.4	50.5, 70.8	53.1, 64.0	63.6, 71.1	

^1^ PP set, pre-protocol set; G#, group no. ^2^
*p* value for Fisher’s exact test (F) for categorical variables and ANOVA (A) or Kruskal–Wallis test (K) for continuous variables. ^3^ NS, non-smoker.

**Table 4 foods-13-04021-t004:** Hematology and blood chemistry tests (Safety Set).

Parameters ^1^	HDPB (*n =* 27)	HDGB (*n =* 28)	PLA (*n =* 28)	*p* Value ^2^
WBC (10^3^/μL)	6.1 ± 1.4	6.2 ± 1.6	6.2 ± 1.6	0.9724 (A)
RBC (10^6^/μL)	4.77 ± 0.49	4.72 ± 0.50	4.8 ± 0.55	0.9528 (A)
Hb (g/dL)	14.4 ± 1.4	14.3 ± 1.4	14.3 ± 1.6	0.9504 (A)
Hct (%)	44.3 ± 3.9	43.9 ± 4.0	44.1 ± 5.0	0.9427 (A)
Platelet (10^3^/μL)	305 ± 51	296 ± 50	295 ± 57	0.7555 (A)
Neutrophil (%)	50.4 ± 9.9	49.7 ± 8.8	48.6 ± 8.3	0.7745 (A)
Lymphocyte (%)	39.3 ± 9.0	39.4 ± 8.2	40.9 ± 7.9	0.7596 (A)
Monocyte (%)	6.7 ± 1.7	6.9 ± 1.6	6.8 ± 1.9	0.9346 (A)
Eosinophil (%)	2.9 ± 2.7	3.4 ± 2.5	3.1 ± 2.7	0.6984 (K)
Basophil (%)	0.66 ± 0.28	0.59 ± 0.22	0.60 ± 0.26	0.6907 (K)
AST (GOT) (U/L)	20.8 ± 6.7	18.2 ± 6.1	19.2 ± 6.6	0.1800 (K)
ALT (GPT) (U/L)	15.2 ± 8.0	14.2 ± 6.3	14.5 ± 5.6	0.8750 (K)
γ-GTP (U/L)	20 ± 16	21 ± 15	20 ± 13	0.8746 (K)
Total protein (g/dL)	7.31 ± 0.35	7.18 ± 0.38	7.20 ± 0.46	0.4895 (A)
BUN (mg/dL)	13.2 ± 2.3	13.6 ± 3.3	13.5 ± 2.6	0.8994 (A)
Creatinine (mg/dL)	0.81 ± 0.15	0.82 ± 0.17	0.81 ± 0.19	0.9311 (K)
Uric acid (mg/dL)	5.8 ± 1.3	5.9 ± 1.4	5.8 ± 1.4	0.9085 (K)
ALP (U/L)	61 ± 16	62 ± 15	62 ± 17	0.9579 (A)
Bilirubin (mg/dL)	0.84 ± 0.28	0.77 ± 0.29	0.77 ± 0.38	0.1693 (K)
Glucose (mg/dL)	74.4 ± 6.1	76.3 ± 6.7	74.4 ± 6.6	0.5971 (K)
Total cholesterol (mg/dL)	194 ± 30	184 ± 27	189 ± 31	0.5344 (A)
HDL cholesterol (mg/dL)	61 ± 15	60 ± 13	61 ± 13	0.8994 (A)
LDL cholesterol (mg/dL)	114 ± 27	108 ± 26	110 ± 28	0.6959 (A)
Triglyceride (mg/dL)	103 ± 49	108 ± 56	112 ± 55	0.7918 (K)

Values are presented as mean ± SD. ^1^ Hematologic tests: white blood cell (WBC), red blood cell (RBC), hemoglobin (Hb), hematocrit (Hct), platelet, neutrophil, lymphocyte, monocyte, eosinophil, and basophil. Blood chemistry tests: AST(GOT), ALT(GPT), γ-GTP, Total protein, Blood Urea Nitrogen (BUN), Creatinine, Uric acid, Alkaline Phosphatase (ALP), Bilirubin, Glucose, Total cholesterol, HDL cholesterol, LDL cholesterol, and triglyceride. ^2^ *p* value for ANOVA (A) or Kruskal–Wallis test (K).

**Table 5 foods-13-04021-t005:** Vital signs (blood pressure and pulse) (Safety set).

Parameters		HDPB (*n =* 27)	HDGB (*n* = 28)	PLA (*n =* 28)	*p* Value ^1^
Systolic blood pressure (mmHg)	Baseline (before ingestion)	117 ± 13	113 ± 11	116 ± 10	0.3578 (K)
	Completed drinking 15 h	118 ± 15	117 ± 13	115 ± 11	
	Change from baseline	1.3 ± 12.7	4.2 ± 10.2	−0.5 ± 11.4	0.3478 (A)
	*p* value ^2^	0.5900	0.0535	0.8118	
Diastolic blood pressure (mmHg)	Baseline (before ingestion)	70 ± 13	69 ± 11	71 ± 11	0.7704 (A)
	Completed drinking 15 h	71 ± 12	70 ± 11	70 ± 10	
	Change from baseline	0.5 ± 10.5	1.0 ± 9.4	−1.9 ± 11.2	0.5604 (A)
	*p* value ^2^	0.8001	0.5994	0.3907	
Pulse (Times/min)	Baseline (before ingestion)	85 ± 12	84 ± 10	86 ± 9	0.6347 (K)
	Completed drinking 15 h	82 ± 15	80 ± 11	80 ± 12	
	Change from baseline	−2.6 ± 12.8	−3.0 ± 11.0	−5.7 ± 9.8	0.5562 (A)
	*p* value ^2^	0.3030	0.1805	0.0060 *	

^1^ *p* value for ANOVA (A) or Kruskal–Wallis test (K). ^2^ *p* value for the paired *t*-test. * *p* < 0.05.

**Table 6 foods-13-04021-t006:** Hourly AHS (points) in the PP set.

Symptoms	HDPB (*n =* 25)	HDGB (*n* = 25)	PLA (*n =* 25)	*p* Value ^1^
Total score	6.7 ± 5.7	8.3 ± 8.8	7.6 ± 6.7	0.3241
Thirst	2.9 ± 1.6	3.2 ± 2.2	3.1 ± 1.7	0.5705
Hangovers	0.80 ± 1.22	0.96 ± 1.14	0.88 ± 1.17	0.7310
Fatigue	1.32 ± 1.55	1.64 ± 1.75	1.52 ± 1.76	0.5204
Headaches	0.76 ± 1.23	0.88 ± 1.64	0.84 ± 1.55	0.8609
Dizziness/Fainting	0.12 ± 0.33	0.36 ± 1.25	0.16 ± 0.47	0.3372
Anorexia	0.12 ± 0.33	0.28 ± 0.61	0.24 ± 0.52	0.4525
Gastrointestinal disorders	0.20 ± 0.50	0.28 ± 0.74	0.52 ± 1.00	0.0152 *
Nausea	0.20 ± 0.58	0.24 ± 0.72	0.24 ± 0.52	0.9163
Heart palpitations	0.28 ± 0.61	0.48 ± 1.16	0.08 ± 0.28	0.0798

Values are presented as mean ± SD. ^1^ Compared within groups; *p* value for RM ANOVA. * *p* < 0.05.

**Table 7 foods-13-04021-t007:** Hourly AHS (points) in the PP set in the GG genotype.

Symptoms	HDPB (*n =* 22)	HDGB (*n* = 22)	PLA (*n =* 22)	*p* Value ^1^
Total score	6.7 ± 5.9	8.2 ± 9.0	7.8 ± 6.9	0.4594
Thirst	2.9 ± 1.5	3.1 ± 2.1	3.2 ± 1.7	0.5828
Hangovers	0.86 ± 1.28	0.95 ± 1.17	0.91 ± 1.19	0.9142
Fatigue	1.41 ± 1.62	1.64 ± 1.81	1.64 ± 1.84	0.7019
Headaches	0.82 ± 1.30	0.91 ± 1.74	0.86 ± 1.61	0.9355
Dizziness/Fainting	0.09 ± 0.29	0.36 ± 1.33	0.18 ± 0.50	0.3628
Anorexia	0.05 ± 0.21	0.27 ± 0.63	0.23 ± 0.53	0.2514
Gastrointestinal disorders	0.18 ± 0.50	0.23 ± 0.69	0.50 ± 1.01	0.0239 *
Nausea	0.18 ± 0.59	0.23 ± 0.75	0.23 ± 0.53	0.9167
Heart palpitations	0.27 ± 0.63	0.45 ± 1.18	0.09 ± 0.29	0.1704

Values are presented as mean ± SD. ^1^ Compared within groups; *p* value for RM ANOVA. * *p* < 0.05.

**Table 8 foods-13-04021-t008:** Compare groups of AUCs over time (PP set).

Variable	Concentration ^1^	HDPB (*n =* 25)	HDGB (*n* = 25)	PLA (*n =* 25)	*p* Value ^2^
Blood alcohol (%)	AUC	0.94 ± 0.19	0.92 ± 0.19	0.89 ± 0.20	0.2764
	iAUC	0.28 ± 0.14	0.26 ± 0.15	0.20 ± 0.16	0.0170 *
	C_max_	0.16 ± 0.02	0.16 ± 0.02	0.16 ± 0.03	0.8374
	T_max_	1.03 ± 0.55	1.06 ± 0.60	0.83 ± 0.66	0.1823

Values are presented as mean ± SD. ^1^ AUC, area under the concentration–time curve; C_max_, maximum plasma concentration; T_max_, time to reach C_max_. ^2^ *p* value for the paired *t*-test (compared between groups). * *p* < 0.05.

## Data Availability

The original contributions presented in this study are included in the article; further inquiries can be directed to the corresponding author.

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
