# Peer review of "Clinical Evaluation of Hovenia dulcis Extract Combinations for Effective Hangover Relief in Humans"

_foods, 2024, doi:10.3390/foods13244021_

Round 1

Reviewer 1 Report

Comments and Suggestions for Authors

1. In this study, the effects of HDPM and HDGM on hangover symptoms in humans were investigated. Results showed that HDPM not HDGM effectively reduced the gastrointestinal disturbance score at 15 h after completion of alcohol drinking. Authors claimed that HD and PL extracts may have a synergistic effect. However, to better understand the benefits of HDPM, the effects of HD and PL extracts should also be investigated individually.

2. The compositions of HD and PL extracts and glutathione yeast extract should be provided.

3. The presentation of Figure 3 is not appropriate. It is not easy to find the significant differences among groups. It needs to be revised and improved.

4. “3.2. Changes in blood alcohol and acetaldehyde levels” should be revised as “3.5. Changes in blood alcohol and acetaldehyde levels”

Author Response

Thank you for your patience and recommendations for strengthening our manuscript (ID: foods - 3354725). We would like to thank the reviewers for their time and expertise in providing critical feedback to make this manuscript suitable for publication. We have revised our manuscript according to the editors’ and reviewers’ comments. In addition to these changes, we have also made substantial revisions to improve this manuscript's style, flow, and clarity. We hope these changes improve the overall quality of this manuscript for publication. We have listed the reviewers’ comments and answered them in sequence.

Reviewer #1:

Comment: 1. Results showed that HDPM not HDGM effectively reduced the gastrointestinal disturbance score at 15 h after completion of alcohol drinking. Authors claimed that HD and PL extracts may have a synergistic effect. However, to better understand the benefits of HDPM, the effects of HD and PL extracts should also be investigated individually.

Response: Prior to conducting this human clinical trial, a preliminary study was conducted in which Sprague-Dawley male rats were given a single dose of the hangover-inducing substance Alcohol (10 mL/kg). In this preliminary study, the effects of hangover relief were evaluated individually in each of the four test groups (HD, PL, and GY) with 10 rats per group, including a control group. Figure S1 shows the results of the experiment, which shows the effects of various extract concentrations on blood acetaldehyde and ethanol levels after alcohol consumption (GY was not requested but added together). This experiment involved administering ethanol, a substance known to induce hangovers, to male Sprague-Dawley rats to assess the hangover-relieving effects of the test substances HD and PL concentrates (next page Figure S1 A-F). This experiment is designed to explore the potential effects of HD and PL using Sprague-Dawley rats rather than human clinical subjects, the data from this part of the study were not included in the manuscript, but a brief preliminary experiment was inserted into ‘section 2.3’ for information (Lines 205-209).

-Hovenia dulcis extract (HD, Figures S1A and S1B):

The HD test substance was administered at doses of 5.6 mg/mL (G2, n =10), 8 mg/mL (G3, n =10), and 16 mg/mL (G4, n =10), and the results were compared with a blank control group (0 mg/mL, G1, n =10). A decrease in blood acetaldehyde levels was observed across all dosage groups. At 0.5 h post-administration, the 8 mg/mL and 16 mg/mL groups showed significant reductions (t-test, p < 0.01). At 1 h, all three dosage groups (5.6, 8, and 16 mg/mL) demonstrated significant reductions (t-test, p < 0.05). A reduction in blood ethanol levels was observed in all dosage groups, with a significant decrease noted in the 16 mg/kg group at 3 h post-administration (t-test, p < 0.05).

-Pueraria lobata extract (PL, Figures S1C and S1D):

The PL test substance was administered at doses of 20 mg/kg, 50 mg/kg, and 100 mg/kg, and the results were compared with a blank control group (0 mg/kg). The results in blood acetaldehyde levels demonstrated that at 3 h post-administration, 20 mg/kg and 50 mg/kg groups showed a statistically significant reduction in acetaldehyde levels. Blood Ethanol Concentrations: Although the reductions were not statistically significant, all dosage groups (20,50, and 100 mg/kg) showed a decreasing trend in ethanol concentration. These findings suggest that, under the conditions of this experiment, PL exhibited hangover-relieving effects by influencing the metabolism of acetaldehyde and ethanol in the blood. Based on these results, it was concluded that the HD exhibited hangover-relieving effects by modulating the metabolic processes of acetaldehyde and ethanol in the blood through pharmacological mechanisms.

-Glutathione yeast extract (GY, Figures S1E and S1F):

In the case of ethanol, the blood concentration at 0.5 h and 40 mg/kg in the yeast extract powder 10 and 40 mg/kg groups decreased statistically significantly compared to the placebo group (p < 0.05).

In the case of acetaldehyde, the blood concentration at 6 and 8 h in the yeast extract powder 20 and 40 mg/kg groups did not show statistical significance but showed a decreasing trend. In the case of ethanol, the blood concentration at 0.5 h and 40 mg/kg in the yeast extract powder 10 and 40 mg/kg groups decreased statistically significantly compared to the placebo group (p < 0.05). In the case of acetaldehyde, the hourly blood concentration at 6 and 8 h in the yeast extract powder 20 and 40 mg/kg dose groups did not show statistical significance but showed a decreasing trend. The above results show that the effect of GY on the relief of hangover symptoms was tested by administering a single dose of the hangover-inducing substance alcohol to male rats. The results of the study confirmed that the test substance acts on the metabolic process of ethanol in the blood and is effective in relieving hangovers.

Figure S1. Ethanol (A, C, E) and acetaldehyde (B, D, F) concentrations of Sprague-Dawley rats (each group n = 10) at each time-point. Values are presented as mean ± SD. Significant difference compared within G1 group; t – test, *p < 0.05., **p < 0.01. A and B: Hovenia dulcis extract (HD); C and D: Pueraria lobata extract (PL); E and F: glutathione yeast extract (GY).

Comment: 2. The compositions of HD and PL extracts and glutathione yeast extract should be provided.

Response A new table (now Table 2) has been inserted to detail the composition of the extracts in each experimental group (Lines 198-203). The table has been renumbered sequentially.

Comment: 3. The presentation of Figure 3 is not appropriate. It is not easy to find the significant differences among groups. It needs to be revised and improved.

Response: As an analysis method for the efficacy evaluation variables, the degree of change in each group was analyzed using RM ANOVA. The description of Figure 3 has been replaced with a comparative analysis between groups using ANOVA as follows. “Values are presented as mean ± SD. Compared within groups; p-value for RM ANOVA. *p < 0.05.” (Lines 433-434).

Comment: 4. “3.2. Changes in blood alcohol and acetaldehyde levels” should be revised as “3.5. Changes in blood alcohol and acetaldehyde levels”

Response: Revised (Lines 401).

Reviewer 2 Report

Comments and Suggestions for Authors

The topic of the article is interesting, the results are preasented clearly. However, the soudness of the results is limited since the studied compounds improved only one out of many symptoms of a hangover- gstrointestinal disorders (Tables No 5 and 6). Another weakness of the results are significant differences among the examined participants. In my opinion the manuscript can be accepted as short communication (or a preliminary study) after a small revison. One point needs to be addressed:

1) The Authors should discuss in a more detail the mechanism of reduction of blood alcohol concentration, and increase of blood acetylaldehyde concentration after 6 hours of alcohol intake (Figure 3A and 3B). It was suggested that it was due to an increased activity of the enzyme alcohol dehydrogenase. On the other hand, it was said that the PL extracts inhibits the dehydrogenase thereby preventing accumulation of acetylaldehyde and oxidative damage of liver cells (Introduction, page 2). This discrepancy should be discussed in a more detail.

Author Response

Reviewer #2: The topic of the article is interesting, the results are preasented clearly. However, the soudness of the results is limited since the studied compounds improved only one out of many symptoms of a hangover- gstrointestinal disorders (Tables No 5 and 6). Another weakness of the results are significant differences among the examined participants. In my opinion the manuscript can be accepted as short communication (or a preliminary study) after a small revison. One point needs to be addressed:

Comment: 1) The Authors should discuss in a more detail the mechanism of reduction of blood alcohol concentration, and increase of blood acetylaldehyde concentration after 6 h of alcohol intake (Figure 3A and 3B). It was suggested that it was due to an increased activity of the enzyme alcohol dehydrogenase.

Response: Revised and inserted explanation of experimental data: Combined with our experimental data, we explained the phenomenon of decreased blood alcohol concentration and increased acetaldehyde concentration 6 h after alcohol intake and linked it to the above mechanism.

Revised the following sentence. “However, the accumulation of acetaldehyde may be due to limited ALDH activity, resulting in slower acetaldehyde clearance” in Section 3.5. (Lines 413-415).

Comment: 2) On the other hand, it was said that the PL extracts inhibits the dehydrogenase thereby preventing accumulation of acetylaldehyde and oxidative damage of liver cells (Introduction, page 2). This discrepancy should be discussed in a more detail.

Response: Revised and inserted (Lines 78-80, 419-419, 554, 556, 636-640)

In the Introduction, we have refined the relevant content to improve this section according to the reviewers' comments. We have added relevant literature citations to provide more support for the proposed mechanism of action of PL in inhibiting (ADH) activity and protecting hepatocytes (Lines 78-80).

“While some studies report its inhibitory effects on ADH activity, others suggest antioxidant properties that may indirectly improve alcohol clearance and protect liver cells from oxidative damage [25].”

  1. Zhao, W.; Peng, D.; Li, W.; Chen, S.; Liu, B.; Huang, P.; Wu, J.; Du, B.; Li, P. Probiotic-fermented Pueraria lobata (Willd.) Ohwi alleviates alcoholic liver injury by enhancing antioxidant defense and modulating gut microbiota. Sci. Food Agric. 2022, 102(11), 5074–5083. https://doi.org/10.1002/jsfa.12049.

The following sentence has been added to Section 3.5, and relevant references have also been added (Lines 419-429).

“Interestingly, it has been reported that PL exerts its effects through a dual role in alcohol metabolism. First, isoflavones, such as puerarin, can inhibit ADH, potentially reducing acetaldehyde accumulation and preventing oxidative damage to liver cells [55]. Second, under certain conditions, PL extract may indirectly enhance alcohol metabolism by reducing oxidative stress and supporting overall enzymatic activity [56]. This apparent discrepancy may be due to differences in experimental models, dosages, or the specific composition of the extracts used. Our results suggest that PL at a concentration of 0.1% (w/v) may exert its effects primarily through antioxidant pathways rather than direct inhibition of ADH. This hypothesis is supported by the observed reduction in blood alcohol levels after 6 h, together with increased acetaldehyde concentrations, which is consistent with a partial enhancement of ADH activity in vivo.”

  1. Lee, K.S.; Kim, G.H.; Seong, B.J.; Kim, H.H.; Kim, M.Y.; Kim, M.R. Effects of aqueous medicinal herb extracts and aqueous fermented extracts on alcohol-metabolizing enzyme activities. Korean J. Food Preserv. 2009,16(2), 259–265.
  2. Feng, R.; Chen, J.-H.; Liu, C.; Xia, F.-B.; Xiao, Z.; Zhang, X.; Wan, J.A combination of Pueraria lobata and Silybum marianum protects against alcoholic liver disease in mice. Phytomedicine 2019, 59, 152824. https://doi.org/10.1016/j.phymed.2019.152824.

The mechanism of action of PL in inhibiting ADH activity and protecting hepatocytes reported in the literature was discussed in detail in section 3.5, and its effect on alcohol metabolism was analysed in combination with experimental data. We have clarified and discussed the differences that may be caused by the experimental conditions, the ratio of extract components and the duration of action, thereby reasonably explaining the dual effects of PL extract, which may inhibit ADH activity and indirectly promote alcohol metabolism.

Reviewer 3 Report

Comments and Suggestions for Authors

This manuscript investigated the potential benefits of HDPM and HDGM in reducing alcohol intoxication and alleviating hangover symptoms. It is interesting and easy to read, suggesting the potential for natural phytochemicals containing HD to accelerate recovery from hangover symptoms. The comments are described below:

1. In the Introduction section, the authors only outlined the characteristics of HD, PL and glutathione-enriched yeast extract. It would be beneficial to discuss the existing studies that have investigated plant extracts in reducing alcohol intoxication and alleviating hangover symptoms. Furthermore, the purpose of this manuscript should be rewritten.

2. Few parameters were measured in this manuscript, including alcohol and acetaldehyde in blood.  The amount of this study is limited. More work, such as Effects of HDPM and HDGM on gut microflora, the protection of HDPM and HDGM on liver tissue, should be carried out.

Author Response

Reviewer #3: This manuscript investigated the potential benefits of HDPM and HDGM in reducing alcohol intoxication and alleviating hangover symptoms. It is interesting and easy to read, suggesting the potential for natural phytochemicals containing HD to accelerate recovery from hangover symptoms. The comments are described below:

Comment: 1. In the Introduction section, the authors only outlined the characteristics of HD, PL and glutathione-enriched yeast extract. It would be beneficial to discuss the existing studies that have investigated plant extracts in reducing alcohol intoxication and alleviating hangover symptoms. Furthermore, the purpose of this manuscript should be rewritten.

Response: Revised the following sentence (Lines 87-93).

“This trial evaluates the efficacy of HD, PL, and GY, in combination to prevent hangover symptoms through a randomized, double-blind, placebo (PLA)-controlled crossover clinical trial. By analyzing blood alcohol and acetaldehyde levels post-alcohol consumption, we aim to provide robust evidence to support the use of these extracts as functional ingredients. This research not only bridges the gap in the clinical validation of natural remedies but also highlights their potential for the development of effective functional foods to mitigate alcohol-related harm.”

Comment: 2. Few parameters were measured in this manuscript, including alcohol and acetaldehyde in blood. The amount of this study is limited. More work, such as Effects of HDPM and HDGM on gut microflora, the protection of HDPM and HDGM on liver tissue, should be carried out.

Response: Removed (Line 146) and revised (Lines 67-70, 335-337, 445, 457, 458) and will be further investigated in future experiments.

Reviewer 4 Report

Comments and Suggestions for Authors

1. The title could be more specific by mentioning the key ingredients. This would increase its accuracy and appeal to specialized readers interested in functional foods.

2. The keywords are relevant but too general. It is recommended that more specific terms be included.

3. The abstract exceeds the 200-word limit established by Foods. It should be more concise, eliminating unnecessary technical details and focusing on the essential findings and their relevance to functional foods.

4. The first paragraph of the introduction lacks elements that capture the reader's attention, such as shocking data or a more interesting connection to the problem. The transition to the study hypothesis could be smoother, following the funnel technique (from general to specific). The study's link to the development of commercial products in functional foods or nutraceuticals is not sufficiently emphasized.

7. The sample size appears limited (25 participants); it needs to be explained in detail how this size was determined to be sufficient to detect significant differences.

8. Details on how adherence to the protocol was ensured, such as abstinence from alcohol before visits, need to be included.

9. Figure 2 and Table 2 contain too many undefined abbreviations, making interpretation difficult. All abbreviations should be explained in the caption or included in a legend.

10. Figure 3 is blurred and not understood, affecting the communication of critical results. It needs to be improved in terms of resolution and clarity.

11. The format of p values is inconsistent. Some tables use three decimal places, while the overall results usually have two. The format needs to be standardized.

12. Details on statistical assumptions, such as normality of data or homogeneity of variances, need to be included, especially in the crossover design. A complete justification of the power analysis and its relation to the sample size must be included.

13. Non-significant results, such as non-gastrointestinal hangover symptoms, are not sufficiently analyzed. Hypotheses should be proposed to explain why some symptoms did not improve.

14. The discussion needs to sufficiently explore how these findings could be applied in developing supplements or functional products.

15. There is no detailed analysis of the formulations' competitiveness in the functional food market or how they could be differentiated from other available products.

16. The conclusions are general and need to sufficiently explore the study's practical implications, such as commercial viability or impact on public health.

17. No concrete proposals for future research are included, limiting this section's depth.

18. The figures have resolution problems, affecting the article's visual quality. It is recommended that they be exported at 300 dpi in high-quality formats such as PNG or TIFF.

19. Excessive use of abbreviations in figures and tables affects the article's clarity. It is necessary to reduce them or make sure they are defined.

Author Response

Reviewer #4: All comments are inserted in the pdf file attached. The quality of the English language is high, though some minor changes are needed.

Comment: 1. The title could be more specific by mentioning the key ingredients. This would increase its accuracy and appeal to specialized readers interested in functional foods.

Response: The title has been changed as follows, taking the author's opinion into consideration Lines (1,2).

“Clinical Evaluation of Hovenia dulcis Extract Combinations for Effective Hangover Relief in Humans”

Comment: 2. The keywords are relevant but too general. It is recommended that more specific terms be included.

Response: Inserted and Changed “Pueraria lobata; acute hangover scale; GC-MS ” (Lines 37, 38).

Comment: 3 The abstract exceeds the 200-word limit established by Foods. It should be more concise, eliminating unnecessary technical details and focusing on the essential findings and their relevance to functional foods.

Response: Revised the following sentence (Lines 22-36). “Alcohol consumption is associated with both short- and long-term adverse effects, including hangover symptoms. The objective of this study was to examine the potential benefits of traditional beverages containing a combination of Hovenia dulcis extract (HD) with either Pueraria lobata extract (HDPM) or glutathione yeast extract (HDGM) in abbreviating alcohol intoxication and mitigating hangover symptoms. A total of 25 participants between the ages of 19 and 40 who had previously experienced a hangover were evaluated in a randomized, double-blind, crossover, placebo (PLA)-controlled clinical trial. Results showed that lower blood alcohol concentrations in the HDPM and HDGM groups were significantly lower than in the PLA group at 0.25 and 0.5 h, suggesting that HD aids in early alcohol metabolism (0 h, p < 0.05). Analysis of the hourly Analysis of the hourly Acute Hangover Scale (AHS) showed that all treatment groups had significantly reduced gastrointestinal disorder symptoms compared to the PLA group (p < 0.05). It can be confirmed that hangover symptoms can be significantly improved by consuming HD combination drinks, apart from the effect of reducing blood alcohol and acetaldehyde concentrations. Therefore, it is predicted that the consumption of natural phytochemicals added to HD is safe for humans and may help accelerate recovery from hangover symptoms.”

Comment: 4. The first paragraph of the introduction lacks elements that capture the reader's attention, such as shocking data or a more interesting connection to the problem. The transition to the study hypothesis could be smoother, following the funnel technique (from general to specific). The study's link to the development of commercial products in functional foods or nutraceuticals is not sufficiently emphasized.

Response: Revised (Lines 41-48).

Comment: 7. The sample size appears limited (25 participants); it needs to be explained in detail how this size was determined to be sufficient to detect significant differences.

Response: The 25 subjects of this experiment believe that the ‘Double-blind Randomised Clinical Evaluation’ is sufficient, and the same number of subjects have also experimented in the Foods Journal below.

Subjects (n = 26): Jeong, I.-K.; Han, A.; Jun, J.E.; Hwang, Y.-C.; Ahn, K.J.; Chung, H.Y.; Kang, B.S.; Choung, S.-Y. A Compound Containing Aldehyde Dehydrogenase Relieves the Effects of Alcohol Consumption and Hangover Symptoms in Healthy Men: An Open-Labeled Comparative Study. Pharmaceuticals 202417, 1087. https://doi.org/10.3390/ph17081087

Subjects (n = 30): Sun, H.; Park, S.; Mok, J.; Seo, J.; Lee, N.D.; Yoo, B. Efficacy and Safety of Wilac L Probiotic Complex Isolated from Kimchi on the Regulation of Alcohol and Acetaldehyde Metabolism in Humans. Foods 202413, 3285. https://doi.org/10.3390/foods13203285

Subjects (n = 24): Lee, S.H.; Choi, S.P.; Park, E.O.; Jung, S.J.; Chae, S.W.; Park, Y.S. Alleviation of hangover effects by DA-5521: A randomized, double-blind, placebo-controlled, crossover trial. Food Eng. Prog. 2024, 28, 20–30. https://doi.org/10.13050/foodengprog.2024.28.1.20.

Comment: 8. Details on how adherence to the protocol was ensured, such as abstinence from alcohol before visits, need to be included.

Response: Included and revised the following sentence (Lines 148-149, 164, 216-218).

“Before randomization at Visit 2 (Day 0) and before the start of the human clinical trial at Visits 3, and 4 (Day 0) to determine alcohol consumption.”

“(2) an exhaled alcohol concentration of 0.00% at the initial visit on Day 2”

“All participants were instructed to abstain from alcohol for 24 h prior to each visit [Visit 1, Visit 2 (Day 1), and Visit 3 (Day 1)]. They were then provided with an identical meal and subsequently ingested either the PLA or the intervention beverage 2 h later.”

Comment: 9. Figure 2 and Table 2 contain too many undefined abbreviations, making interpretation difficult. All abbreviations should be explained in the caption or included in a legend.

Response: Revised. Abbreviations are shown in the legend of Figure 2 and Table (now Table 3) (Lines 301-303, 323).

Comment: 10. Figure 3 is blurred and not understood, affecting the communication of critical results. It needs to be improved in terms of resolution and clarity.

Response: High-resolution (300 dpi) Figure 3 have been replaced (Line 429).

Comment: 11. The format of p values is inconsistent. Some tables use three decimal places, while the overall results usually have two. The format needs to be standardized.

Response: Revised; all p-values were shown as four decimal places in Tables. However, in the abstract and conclusion sections and at the foot under the Tables 5-8, values with p-values of 0.05 or less were consistently corrected by displaying ‘*p < 0.05’ (Lines 356, 386, 389, 399).

Comment: 12. Details on statistical assumptions, such as normality of data or homogeneity of variances, need to be included, especially in the crossover design. A complete justification of the power analysis and its relation to the sample size must be included.

Response: In this study, the degree of change in each group was analysed using RM ANOVA as the analysis method for efficacy evaluation variables, and in addition, an ANOVA was also performed to identify differences in the intake effects between each group under the assumption that there was no residual effect, and to assess whether there was a statistically significant difference. As for the analysis method for continuous variables, the normality test (Shapiro-Wilk test) was performed to compare groups, and ANOVA was used when normality was satisfied in all groups based on a p-value of 0.05. If normality was not satisfied in even one group, a non-parametric alternative such as the Kruskal-Wallis test was used to ensure the validity of the results. In cross-over designs, the potential carry-over effect was tested by including interaction terms in the model. In addition, each subject was used as a control group to minimise the variability of the study and to increase statistical power. Confirmatory trials were used to inform the design of efficacy studies, such as calculating the optimal dose and number of samples with adequate power. In this human application trial, we sought to confirm the clinical efficacy and safety of the Hovenia dulcis extract combinations in an exploratory manner, and we enrolled a total of 30 human study subjects, taking into account the minimum statistical requirements and potential dropouts. This was sufficient to maintain statistical power and ensure reliable results. We have included these details in the text to increase transparency and fully justify our methodological choices (Lines 111, 112, 268-277).

“The trial used a crossover design with participants randomized to one of six groups, with a target of 30 participants to allow for a potential 20% dropout rate.”

Comment: 13. Non-significant results, such as non-gastrointestinal hangover symptoms, are not sufficiently analyzed. Hypotheses should be proposed to explain why some symptoms did not improve.

Response: The results of the nine-item AHS, measured 15 h after alcohol consumption, showed a tendency for HDPB to have lower scores than PLA on all AHS items overall, but no significant differences were found on items other than 'gastrointestinal disturbance'. When designing the human study, we expected that the greater the amount of water, the less the deviation in test score values. It is well known that even adequate water intake helps to improve hangovers, and in the case of this test it is speculated that some symptoms were not improved due to the water content of the test product, which was 500 mL. The following sentence was inserted into the manuscript.

Added “Although no significant differences were observed in other measures, the scores of the HDPM group were consistently lower than those of the PLA group. This indirectly demonstrates the efficacy of the mixed extract of HD and PL in alleviating hangover symptoms.” (Lines 380-383)

Comment: 14. The discussion needs to sufficiently explore how these findings could be applied in developing supplements or functional products.

Response: When developing functional food products that relieve hangovers, the impact of water content can be considered in the product volume design and blood alcohol absorption rate. The conclusion has been inserted and modified (Lines 459-467).

Comment: 15. There is no detailed analysis of the formulations' competitiveness in the functional food market or how they could be differentiated from other available products.

Response: There is no other case in the functional food market where a human application test for hangover relief has been conducted with a test group (500 mL) with a high-water content as in the current test (currently less than 100 mL). In addition, unlike other tests, the test subjects were not given either snacks or fluids during or after drinking. Therefore, the results of the test product could be seen more clearly. HDPB was shown to be more effective than PLA in improving hangover symptoms after the test (Lines 459-467).

Comment: 16. The conclusions are general and need to sufficiently explore the study's practical implications, such as commercial viability or impact on public health.

Response: In general, drinking enough fluids can also help alleviate some of the symptoms of a hangover by addressing the dehydration caused by alcohol consumption. In addition, there are no studies to date that have compared the effects of drinking enough water (500 mL of beverage) containing a plant-based concentrated extract that has been shown to help relieve hangovers in human trials, so the results of this study are considered to have commercial and academic value in terms of public health impact (Lines 459-467).

Comment: 17. No concrete proposals for future research are included, limiting this section's depth.

Response: For the synergistic combination/analysis of HD and PL ingredients, the extract-mixture ingredients themselves will be analysed using various techniques (DPPH, ADH, ALDH, AST, ALT, gamma GTP, effects on the intestinal microbiome, etc.) to determine the mechanism of hangover relief (Lines 457, 459-467).

Comment: 18. The figures have resolution problems, affecting the article's visual quality. It is recommended that they be exported at 300 dpi in high-quality formats such as PNG or TIFF.

Response: High-resolution (300 dpi) Figure 3 have been replaced (Line 429).

Comment: 19. Excessive use of abbreviations in figures and tables affects the article's clarity. It is necessary to reduce them or make sure they are defined.

Response: The abbreviations in the figures and tables have been modified to aid in the clarity of the manuscript (Lines 199, 200, 322, 340).

Round 2

Reviewer 1 Report

Comments and Suggestions for Authors

The authors have adequately addressed my comments and suggestions for revision.

Reviewer 3 Report

Comments and Suggestions for Authors

The manuscript is improved significantly, and can be considered to be published.

Reviewer 4 Report

Comments and Suggestions for Authors

Accept in present form.